# Congenital Fibrosis of the Extraocular Muscles: An Overview from Genetics to Management

**DOI:** 10.3390/children9111605

**Published:** 2022-10-22

**Authors:** Weiyi Xia, Yan Wei, Lianqun Wu, Chen Zhao

**Affiliations:** 1Department of Ophthalmology and Vision Science, Eye & ENT Hospital, Shanghai Medical School, Fudan University, 83 Fenyang Road, Shanghai 200031, China; 2Key Laboratory of Myopia, Ministry of Health, Fudan University, 83 Fenyang Road, Shanghai 200031, China; 3Shanghai Key Laboratory of Visual Impairment and Restoration, Fudan University, 83 Fenyang Road, Shanghai 200031, China

**Keywords:** congenital cranial dysinnervation disorders, congenital fibrosis of the extraocular muscles, genotype–phenotype correlation, classification, management

## Abstract

Congenital fibrosis of the extraocular muscles (CFEOM) is a genetic disorder belonging to the congenital cranial dysinnervation disorders and is characterized by nonprogressive restrictive ophthalmoplegia. It is phenotypically and genotypically heterogeneous. At least seven causative genes and one locus are responsible for the five subtypes, named CFEOM-1 to CFEOM-5. This review summarizes the currently available molecular genetic findings and genotype–phenotype correlations, as well as the advances in the management of CFEOM. We propose that the classification of the disorder could be optimized to provide better guidance for clinical interventions. Finally, we discuss the future of genetic-diagnosis-directed studies to better understand such axon guidance disorders.

## 1. Introduction

Congenital fibrosis of the extraocular muscles (CFEOM) refers to a subset of congenital cranial dysinnervation disorders (CCDDs). It is characterized by hereditary nonprogressive restrictive ophthalmoplegia, with or without blepharoptosis [1]. The estimated prevalence of this complex is not less than 1/230,000 [2]. The most distinctive feature of ophthalmoplegia is that vertical eye movements are severely restricted, commonly resulting in a compensatory chin-up head position or head tilt [3]. Amblyopia is also frequently observed in CFEOM. However, recent studies by Thomas et al. and Khan et al. noted frequent optic nerve hypoplasia and retinal changes in patients, suggesting that subnormal vision is not fully ascribable to amblyopia [4,5]. These findings explain that the subnormal vision is not fully ascribable to amblyopia and expand the phenotypic spectrum of CFEOM beyond ophthalmoplegia. Neurological examinations typically reveal hypoplasia or aplasia of the oculomotor nerve, with or without the absence of the trochlear nerve and levator palpebrae superioris muscles [6,7]. This review summarizes the currently available molecular genetic findings and genotype–phenotype correlations, as well as the advances in the management of CFEOM. We also explore the possibility of taking advantage of classification theory and genetic diagnoses to manage CFEOM.

## 2. CFEOM Subtypes

Five subtypes of CFEOM have been established: CFEOM-1 (MIM #135700), CFEOM-2 (MIM #602078), CFEOM-3 (MIM #600638), CFEOM-4 (OMIM #609428), and CFEOM-5 (OMIM #616219). CFEOM-1 is the classic subtype, which is typically characterized by bilateral restrictive ophthalmoplegia, with globes partially or fully fixed downward, the inability of supraduction above the horizontal midline, and bilateral blepharoptosis (Figure 1) [8]. Marcus Gunn jaw winking has been observed in some CFEOM-1 patients and is considered a feature of this subtype [1]. CFEOM-2 typically presents with bilateral restrictive ophthalmoplegia with eyes partially or completely fixed in a considerably exotropic position, pupil abnormalities, and blepharoptosis (Figure 2) [9]. This subtype is also frequently associated with the atrophy of the superior oblique muscle through neuroimaging [10]. The clinical features of CFEOM-3 are variable, including bilateral asymmetry, varying degrees of ophthalmoplegia and blepharoptosis, from mild to complete, and varying eye positions at the primary gaze (Figure 3) [11]. Affected individuals frequently present additional neurological signs and symptoms in addition to mental abnormalities in CFEOM-3 [12]. Recently, Thomas et al. suggested that some patients diagnosed with congenital monocular elevation deficiency could be included in CFEOM-3 [13]. CFEOM-4, also known as Tukel syndrome, is characterized by a CFEOM-3 phenotype plus hand oligodactyly [14,15]. Recently, CFEOM-5 was proposed by Munezane et al. as a novel subtype, characterized by paralytic exotropia, congenital ptosis, and/or globe retraction with synergistic divergence [16,17]. Regardless, the specific ophthalmic phenotypes of CFEOM-5 have yet to be detailed. Although CFEOM-1 is the most common phenotype worldwide, the incidences of these five subtypes vary among races. For instance, Chen et al. reviewed 40 Chinese cases of CFEOM and found that 29 cases (72.5%) were CFEOM-1, 10 cases (25%) were CFEOM-3, and only 1 case (2.5%) was CFEOM-2 [18]. This is consistent with the spectrum in Western countries, whereas there is a significantly higher incidence of CFEOM-2 in the Middle East due to consanguineous marriages [19,20,21,22,23].

## 3. Genetics

CFEOM is a monogenic disorder with significant genetic heterogeneity. To date, seven disease-causing genes (*KIF21A*, *PHOX2A, TUBB3, TUBB2B, TUBA1A*, *ECEL1*, and COL25A1) and one locus have been described as correlating with CFEOM. Among them, *KIF21A*, *PHOX2A*, and *TUBB3* have been recognized for a relatively long time: over two decades. By contrast, mutations in *TUBB2B, TUBA1A*, *ECEL1*, and *COL25A1* that cause CFEOM phenotypes were only recently reported in sporadic cases. Pathogenic variants in these genes generally result in the abnormal development of either cranial nuclei or nerve axons connected to the extraocular muscles, which is also known as “dysinnervation” [24]. It is now accepted that such primary dysinnervation leads to the fibrosis of the extraocular muscles and consequent ophthalmoplegia. However, the specific pathogenic mechanisms underlying secondary fibrosis remain to be elucidated. Meanwhile, CFEOM exhibits clinical heterogeneity, depending on the origin and severity of the dysinnervation. Regardless, the clinical phenotype differences among the five forms reflect the differences in the functions of the mutated genes. Molecular genetic diagnoses, in turn, provide valuable information on the natural history, including systemic manifestations and ocular presentations. Table 1 summarizes the associated genes and the corresponding clinical findings.

### 3.1. KIF21A

*KIF21A* (OMIM #608283) encodes a microtubule-associated kinesin-encoding protein. Mutations in this gene are the most common cause of CFEOM worldwide, accounting for over half of all cases. These mutations cause almost all CFEOM-1 cases and a small proportion of CFEOM-3 cases. The corresponding condition is autosomal dominantly inherited with complete penetrance, although the germline mosaicism that occurs in *KIF21A*-related CFEOM can be disguised as a recessive inheritance pattern [8,25]. *KIF21A* variants seldom cause peripheral sensorimotor neuropathy or central nervous system (CNS) malformations [26]. KIF21A belongs to the molecular motor kinesin family, comprising an N-terminal motor domain, three central coiled-coil stalk domains, and a C-terminal tail domain containing seven WD40 repeats (Figure 4). To date, at least 15 *KIF21A* pathogenic variants and related amino acid substitutions accounting for the pathomechanisms have been identified, influencing ocular motor neuron axonal guidance [25,27,28,29,30,31,32,33,34,35,36]. Remarkably, the majority of these amino acid substitutions were located in the second coiled-coil stalk domain according to UniProt predictions [37]. Functional studies supported the notion that substitutions within this domain substantially upregulate KIF21A activity by diminishing the autoinhibition of the motor and disturbing axon growth toward the oculomotor muscles [38,39]. These findings indicate a gain-of-function mechanism for *KIF21A* variants and emphasize the importance of the second coiled-coil stalk domain in the etiological elucidation of KIF21A-associated CFEOM [40].

### 3.2. PHOX2A

*PHOX2A* (OMIM #602753), also known as *ARIX*, is the only known disease gene associated with CFEOM-2. This condition shows autosomal recessive inheritance with complete penetrance. The gene encodes a homeodomain transcription factor that is essential for the development and survival of oculomotor and trochlear motor neurons. Therefore, oculomotor or trochlear nerves are commonly undetectable by magnetic resonance imaging in affected individuals [41]. To date, five *PHOX2A* mutations have been recognized as causing lower motor neuron sequela in the brainstem and subsequent CFEOM-2, including three missense variants and two nucleotide changes adjacent to the splice site (Figure 5) [9,41,42,43]. Moreover, *PHOX2A* polymorphisms also seem to play a role in congenital superior oblique palsy, in which the superior oblique is hypoplastic but the oculomotor nerve is hardly affected [44,45]. Intriguingly, all of these polymorphisms were located before 200 bp in the *PHOX2A* locus, while all the nucleic acid site changes known to cause CFEOM were located after 200 bp. Pupil abnormalities have been documented in patients with *PHOX2A* mutations; these individuals can be distinguished from individuals of other subtypes. The pupils are usually small but sometimes vary in size and shape. The pupillary reaction is sluggish to both light and accommodation but can be partially induced by pharmacological agents somehow due to iris innervation [41].

### 3.3. TUBB3

Microtubules are assembled from α- and β-tubulin heterodimers. *TUBB3* (OMIM #602661) belongs to a group of tubulin genes and encodes a neuron-specific β-tubulin isotype that is critical for the formation and function of microtubules, as well as for further neuronal migration and axon guidance [11,46]. CFEOM patients harboring *TUBB3* mutations belong to the CFEOM-3 or CFEOM-1 subtype and tend to exhibit significant asymmetry, variable clinical presentations, and incomplete dominance. They frequently have other systemic abnormalities, including intellectual disability, developmental delay, autism spectrum disorder, mild brain malformations, peripheral axonal neuropathy, microphallus, and/or cryptorchidism [47]. In brief, individuals with unilateral CFEOM or CFEOM without ptosis, or individuals manifesting CFEOM-3 as well as neurological deformities, are most likely to bear a *TUBB3* mutation [48]. To date, 13 amino acid substitutions in TUBB3 have been recognized as underlying the CFEOM-3 phenotype (Figure 6), with some displaying incomplete penetrance (e.g., R262H and R262C) [11,13,47,49,50,51,52]. Meanwhile, a distinct set of pathogenic *TUBB3* missense variants have been associated with malformations of cortical development (MCD) [51]. CFEOM-3 and MCD were regarded as distinct disorders because TUBB3 variants have distinct functional consequences that resulted in either of these disorders. Missense mutations at specific β-tubulin residues mostly cause CFEOM, and others lead to MCD, although there were some exceptions [11,49,53]. However, two amino acid substitutions in TUBB3 were revealed to cause both CFEOM-3 and MCD phenotypes [49]. Another substitution (G98S), previously known to cause CFEOM-3, was identified in an MCD pedigree, which, however, did not meet the diagnostic criteria for COFEM [54]. Interestingly, recent studies have demonstrated that infantile nystagmus could arise without CFEOM owing to TUBB3 variants [55]. In particular, there are two distinct clinical disorders caused by two specific mutations, E410K and R262H, which can be recognized by their respective hallmarks. Besides ophthalmoplegia and oculomotor hypoplasia, the distinctive key features of TUBB3 E410K syndrome comprise anosmia, hypogonadotropic hypogonadism, pyloric stenosis, and cyclic vomiting [56]. Regarding the TUBB3 R262H syndrome, the clinically recognizable phenotype includes congenital joint contracture, early onset peripheral neuropathy, an abnormal gait, and dysmorphic basal ganglia [50].

### 3.4. TUBB2B and TUBA1A

*TUBB2B* (OMIM #612850) and *TUBA1A* (OMIM #602529) encode a β-tubulin isotype and an α-tubulin isotype, respectively. However, the frequency of CFEOM-causing mutations appears to be much lower than that of *TUBB3*. This is probably due to the fact that mutations in *TUBB2B* or *TUBA1A* were generally more deleterious than those in *TUBB3*, bringing about a higher incidence of embryonic lethality [57]. To date, only one and three heterozygous missense mutations in *TUBB2B* and *TUBA1A*, respectively, have been identified in CFEOM pedigrees, with or without MCD [57,58]. As expected, some of the affected individuals also had neurological malformations. The only implicated pathogenic TUBB2B amino acid substitution, E421K, was revealed to lead to primary axonal dysinnervation and a consequent CFEOM-3 phenotype in a dominant manner. All three *TUBA1A* mutations (H406D, R156H, and M398R) predicted to be detrimental in three unrelated probands were reported in the same research by Jurgens et al. in 2021, causing either CFEOM-1 or CFEOM-3 [58]. Protein structural modeling suggests that these variants affected either the assembly of microtubules or interactions with the motor domain of kinesin-1. Remarkably, another study on cerebellar dysplasia described that some patients harboring *TUBA1A* variants displayed characteristics such as strabismus, ptosis, or unilateral ocular elevation deficiency, while more detailed research is warranted to decide whether these cases met the diagnostic criteria for CFEOM [59]. Meanwhile, a distinct set of *TUBB2B* and *TUBA1A* mutations have also been implicated in neurodevelopmental disorders, including MCD. These findings broaden our knowledge of tubulinopathies and suggest that tubulin-encoding genes should be addressed in CFEOM patients with negative genetic screening results for *KIF21A* and *TUBB3*.

### 3.5. ECEL1

*ECEL1* (OMIM #605896) encodes a type II integral transmembrane zinc metalloprotease predominantly expressed in neurons. It is associated with neuronal development and neuromuscular junction formation [60]. A series of autosomal recessive mutations in *ECEL1* have been documented to cause arthrogryposis multiplex congenita (AMC), a common type of congenital contracture disorder [61]. Additionally, some sporadic cases were recorded with ocular phenotypes in addition to limb phenotypes, including ophthalmoplegia, strabismus, and ptosis, in accordance with a definition of CFEOM [62,63]. Notably, these *ECEL1* mutations exhibited variable penetrance and expressivity; therefore, it is hard to categorize the corresponding phenotypes into specific delineated subtypes. For example, Shaaban et al. identified two patients in a family harboring a c.1819G > A missense mutation with an ocular phenotype of ophthalmoplegia and significant refractive errors, who both met the diagnostic criteria for COFEM [62]. In another study, a c.2278C > T missense mutation was documented to cause ophthalmoplegia and ptosis in an incomplete penetrant manner [64]. Accordingly, these ECEL1-related orbital dysinnervation manifestations substantiated the theory that *ECEL1* is a new responsible gene for CFEOM.

### 3.6. COL25A1

*COL25A1* (OMIM #610004) encodes a transmembrane-type collagen, type XXV, alpha 1, which is essential for intramuscular motor innervation. At present, five recessive pathogenic variants in *COL25A1* have been identified in families with CFEOM-5, including three missense mutations (c.1144G > A, c.1450A > G, and c.1198G > A) and two compound heterozygous mutations (c.672 + 1del plus c.672 + 1G > A, and c.1489G > T plus the deletion of chromosome 4: 109,852,901–109,976,457) [65,66,67]. Patients harboring the c.1144G > A missense mutation, the c.1450A > G missense mutation, or the c.1489G > T plus the deletion of chromosome 4: 109,852,901–109,976,457 compound heterozygous mutation were characterized by globe retraction with synergistic divergence. On the other hand, patients harboring the c.1198G > A missense mutation or the c.672 + 1del plus c.672 + 1G > A compound heterozygous mutation typically presented with incomitant large-angle exotropia and limited adduction. Distinct from the aforementioned genes predominantly expressed in neurons, *COL25A1* is the first gene whose expression in muscles may be accountable for the pathogenesis. Further functional studies implied that these variants significantly reduced the expression of the COL25A1 protein and impaired its interaction with receptor protein tyrosine phosphatases (PTPs) σ and δ [16]. This consequently resulted in altered levels of molecules involved in axonal guidance, such as sAPP and TUBB3, as well as the attenuation of motor axon contacts [66].

## 4. Management

### 4.1. Initial Evaluation and Routine Monitoring

Initially, the medical history should include a family history of similar conditions, ocular surgical history, and other significant systemic histories. A photographic profile should be archived for future comparisons. One important thing to keep in mind is that CFEOM is congenital and nonprogressive, distinguished from chronic progressive external ophthalmoplegia (CPEO). General ophthalmological and orthoptic examinations are required, including slit-lamp examination, pupil examination, dilated fundoscopy, refraction, best-corrected visual acuity (BCVA), palpebral fissure size measurement, levator function assessment, deviation angle determination, ocular motility assessment, and finding evidence of dysinnervation, such as synergistic convergence, Marcus Gunn jaw winking, and abnormal head posture. Special examinations for evaluating the retinal function are optional and could include fundus photography, optical coherence tomography (OCT), visual field assessment, and electroretinography [4]. Even in the absence of other neurologic signs or symptoms, brain and orbital MRIs are performed to determine the absence, hypoplasia, or misdirection of cranial nerves III, IV, and VI, as well as the atrophy of the extraocular muscles.

Although the confirmation of the underlying genotype seems to be less informative for surgical planning in CFEOM, genetic counseling is recommended, particularly for patients with unknown inheritance patterns or who have birth plans. For example, a patient harboring *TUBB3* mutations may have an unremarkable family history because of incomplete penetrance or de novo mutations [48]. To say the least, the genetic approach is the ultimate distinguishing tool to confirm the clinical diagnosis [68]. In particular, Khan et al. [69] recently suggested conducting genetic testing for a CFEOM-like phenotype before general anesthesia to exclude pathogenic mutations in *RYR1* (OMIM #180901), which was documented to be associated with congenital ophthalmoplegia, ptosis with facial weakness, and malignant hyperthermia [70,71].

Regarding routine monitoring, Whitman et al. recommended annual or biannual ophthalmologic examinations to prevent complications arising from corneal exposure [17]. Younger patients at risk of amblyopia are additionally referred to eye care professionals throughout the critical period of vision development three or four times annually.

### 4.2. Surgical Management

Surgical intervention for CFEOM is challenging and technically difficult. Patients with indications for surgery, or their guardians, should be informed of the uncertainty of the surgical outcome, which can include limited or even no improvement in eye movements. The restoration of binocular vision is not a realistic goal [3]. They also need to know that the patient may need to undergo multiple procedures to obtain relatively satisfactory cosmetic outcomes. In a cohort of 13 genetically defined patients, an average of 2.8 strabismus procedures were performed [72]. Younger patients tend to undergo fewer procedures and have better surgical outcomes, partially because the orbital tissues contracture over time. Sener et al. reviewed the surgical outcomes of 52 CFEOM patients and concluded that those with CFEOM-1 had a better prognosis with regard to head position and ocular alignment postsurgery [23]. Commonly due to the weakness of facial muscles, the ocular surfaces of CFEOM-3 patients are more vulnerable; therefore, ophthalmologists need to be more vigilant to evidence of corneal exposure [73].

Orbital MRI is strongly recommended prior to surgery for the detection of abnormalities or even the absence of extraocular muscle(s) and the superior rectus–levator complex [74]. The surgical treatment of CFEOM includes extraocular muscle surgery as well as eyelid repair. Accurate classification by phenotype or genotype is essential for surgical planning. However, the surgical approach should be tailored based on the variable clinical presentations of CFEOM, even within a family with the same molecular defect. The surgeon should consider a number of factors. There should be a “Plan B” to cope with different forced duction test results and possible anatomic variations. The likelihood of stepwise rectus muscle surgery, to avoid anterior segment ischemia, should be considered. The presence of postoperative diplopia in rare patients with gross stereopsis, aberrant movements of synergistic convergence and divergence, the type of pattern deviation, and abnormal head position need to be kept in mind as well [23].

Sequentially, strabismus surgery should take precedence over eyelid repair because head position, eyelid position, and/or Bell’s phenomenon are expected to be altered after rectus muscle surgery [14,75]. Tawfik et al. reported that a patient exhibited eyelid retraction and lagophthalmos with significant corneal exposure after levator resection surgery because the eyelid surgeon did not recognize the manifestations of CFEOM and treated the patient as a “simple congenital ptosis” case [76]. As a result, the patient still had to tilt their head to compensate for the downward fixed eyes after eyelid surgery. Heidary et al. commonly performed eyelid surgery within 2 months of strabismus surgery or even simultaneously, given the considerations of fewer anesthetic administrations and lower travel expenses for families [72].

To identify which extraocular muscles are fibrosed, a forced duction test should be performed prior to surgery. To exclude additional restrictive factors from the eye globe, we regularly implement forced duction again after the rectus muscle has been disinserted. Fibrotic bands and adhesions should be released if present. The procedures for weakening the “frozen” rectus muscle, including recession, hangback sutures, and simple disinsertion, remain the mainstay for relieving restriction. Standard recession amount tables are not suitable in most cases, because the laws of innervation are contravened. Sener et al. mentioned that the rectus muscles weakened by hangback sutures tend to regress and reattach to the original insertion [23]. Therefore, conventional strabismus surgery is commonly insufficient in cases with a large-angle deviation. To maximize the weakening effects, Hedergott et al. suggested tendon elongation techniques using bovine pericardium [77].

Severely affected individuals have eyes fixed in a downward primary position. Improvement in hypotropia predominantly relies on a large/supramaximal inferior rectus recession, from 6 to 12 mm, with or without the application of an adjustable suture. We avoid employing superior rectus resection because the superior rectus muscles commonly receive no innervation, and resection will not yield long-lasting results [78]. Shoshany et al. reported that superior oblique tenotomies were helpful in ameliorating upgaze limitations and anomalous head postures for patients with a nasally displaced insertion of the superior oblique muscles [21]. In addition, for those cases in which the horizontal rectus muscles are mildly affected, upward horizontal rectus muscle transposition is also an option [79,80]. Although dissociated vertical deviation (DVD) rarely coexists, it can be managed with superior rectus recession [81].

If horizontal deviation, usually exotropia, is present, the recession of the restrictive rectus muscle is generally performed and is satisfactory in patients with good adduction. However, the feasibility of ipsilateral antagonist resection remains controversial. In our experience, even if there is no antagonist function, it does help to improve horizontal ocular alignment and to expand the motility range in some cases.

Unlike concomitant strabismus cases, unpredictable postoperative outcomes occasionally occur in patients with CFEOM, mostly due to innervation abnormalities. Patients may exhibit a new horizontal misalignment after vertical strabismus surgery [82]. Even in those patients who obtain excellent primary position alignment and improved head posture, aberrant innervation remains, manifesting as a disfiguring upshot/downshot. Many patients have residual exotropia despite profound weakening procedures [19]. Heidary et al. recommended the use of botulinum toxin as adjunctive therapy for patients with residual misalignment [72].

In terms of ptosis repair, the preoperative assessment of the facial muscle function and corneal sensitivity should routinely be included. Depending on the residual levator function, levator resection or frontalis suspension is chosen for the treatment of ptosis. Blepharoplasty is often performed alongside ptosis surgery. The aim of the operation is to elevate the superior lid margin to 1–2 mm above the pupil in the primary gaze, while the Bell phenomenon should also be taken into account when deciding on the extent of lid surgery [19]. It has also been suggested that temporary materials should be used, or that a brow suspension should simultaneously be performed to reduce the risk of corneal exposure. Eye lubrication is routinely applied after ptosis surgery.

### 4.3. Nonsurgical Management

Nonsurgical management consists of the correction of refractive error, the treatment of amblyopia, and the protection of the ocular surface. Contact lenses are superior to spectacles in the management of spectacle refractive error, as most patients can hardly adapt to spectacles because their abnormal head posture disposes of the optical center [3].

As an additive result of unilateral strabismus, ptosis-induced occlusion (deprivation), and frequent refractive error, there is a high rate of amblyopia in patients with CFEOM. Early intervention for amblyopia has shown a good response; therefore, it is desirable to begin as early as possible. It is also worth noting that extraocular muscle surgery is expected to improve visual acuity in younger children [83,84].

Corneal lubrication is administered for the care of the ocular surface. Prosthetic replacement of the ocular surface ecosystem (PROSE) treatment is optional for complex cases with accessible exposure keratopathy [72].

## 5. Prospects: Optimizing Decisions upon Molecular Genetic Diagnoses

As significant advancements have been made in elucidating the disease-causing genes of CFEOM, the definitions or classifications of CFEOM and CCDDs are evolving. The current classification of CFEOM is based on both clinical presentation and genetic differences and has several limitations. First, the grading of eye movement, from normal to mild and moderate to severe, is limited and is a somewhat subjective judgment that could differ among observers. Second, there is an extensive overlap of clinical manifestations between CFEOM-1 and CFEOM-2, making subtyping confusing sometimes. Third, unlike CFEOM-1 or CFEOM-2, which are caused by mutations in a single gene, CFEOM-3 is less well-defined but more genetically heterogeneous, as mentioned above (Section 3). In addition, given its phenotypic variability, CFEOM-3 is an exclusive diagnosis to some extent. Doherty et al. reported a pedigree in which some affected individuals manifested as CFEOM-1, whereas others manifested as CFEOM-3, making the subtyping confusing [85]. Fourth, given the phenotype overlap between CFEOM-5 and Duane retraction syndrome (DRS), another CCDD subset, whether *COL25A1*-related ocular abnormalities can be subtyped as CFEOM-5 remains to be discussed [66]. Last but not least, clinicians currently receive limited guidance for treatment strategies from this classic classification system; clinical presentations remain the final arbiter of the surgical approach [86].

Eminently, Lee et al. classified DRS into Exo-, Ortho-, and Eso-DRS subtypes according to the horizontal deviation angle at the primary gaze [87]. Compared with the traditional Huber classification for DRS, their new classification system is more convenient and practically useful in clinical settings. An analogous novel CFEOM classification approach that is mainly based on objective indicators is required. The following factors can be considered: the direction of strabismus at the primary gaze; the severity of ptosis; retinal abnormalities; the involvement of the pupil, optic nerve head, and other eye structures; CNS malformations beyond cranial nerves III, IV, and VI; and other systemic manifestations. One of the possible solutions is to classify CFEOM (defined as “congenital nonprogressive restrictive ophthalmoplegia with or without ptosis”) into six types purely according to genotype, regardless of the clinical manifestations. To be specific, patients harboring *KIF21A* mutations were classified as CFEOM-1, patients harboring *PHOX2A* mutations were classified as CFEOM-2, patients harboring mutations in tubulin-encoding genes (*TUBB3*, *TUBB2B*, or *TUBA1A*) were classified as CFEOM-3, patients harboring mutations in the locus of chromosome 21qter (with simultaneous oligodactyly) were classified as CFEOM-4, patients harboring mutations in *COL25A1* were classified as CFEOM-5, and patients harboring mutations in *ECEL1* were classified as CFEOM-6. To say the least, such a genetic classification helps clinicians to know prognoses, give procreation guidance, and prompt prospective care in terms of systemic abnormalities.

CFEOM is a genetic disorder related to the development or functions of motor neurons, and it can influence the physical and mental health of patients if left untreated. Current clinical studies on CFEOM focus mainly on surgical interventions for strabismus; however, the disease is not limited to strabismus. Further studies on how to take advantage of an underlying genetic diagnosis to inform clinical interventions are required [69]. Regarding the expansion of the molecular basis of CFEOM, we suggest that microtubule-associated genes, especially those that encode α- or β-tubulin isotypes, should be included in genetic testing panels for CFEOM and CCDDs. Future functional analyses would provide insights into the precise mechanisms of why specific CFEOM-causing gene mutations, ubiquitously expressed in neurons, predominantly affect the eyelid and ocular motility rather than cause obvious multiorgan disease or severe neurodevelopmental disorders.

## Figures and Tables

**Figure 1 children-09-01605-f001:**
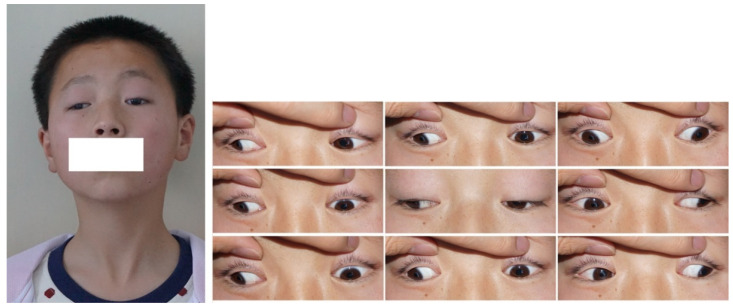
Images of a CFEOM-1 patient with bilateral ptosis, large fixed hypotropia, and a chin-up head posture.

**Figure 2 children-09-01605-f002:**
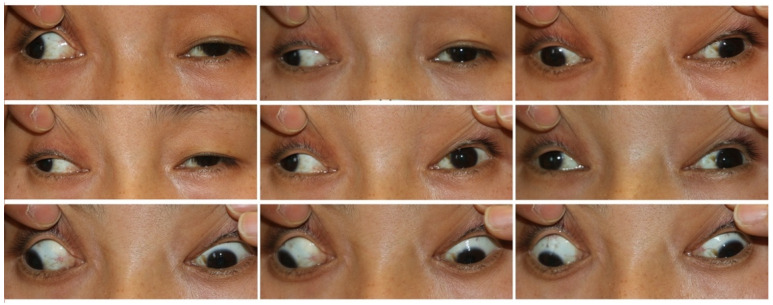
Images of a CFEOM-2 patient with bilateral ptosis and large fixed exotropia.

**Figure 3 children-09-01605-f003:**
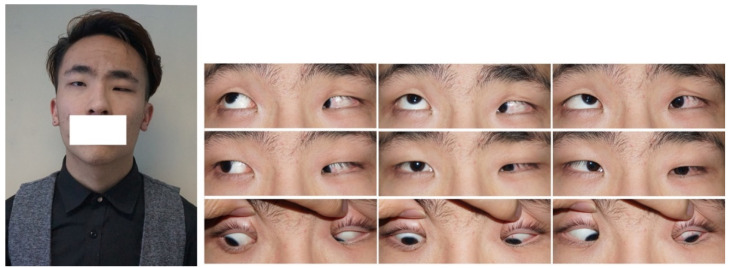
Images of a CFEOM-3 patient with unilateral ptosis, hypotropia, and limited abduction as well as supraduction of the left eye.

**Figure 4 children-09-01605-f004:**
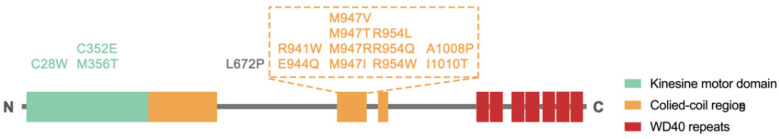
Schematic representation of the KIF21A protein structure and 15 previously recognized disease-causing amino acid substitutions associated with CFEOM.

**Figure 5 children-09-01605-f005:**
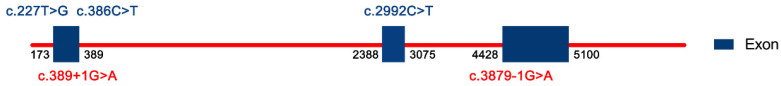
Schematic representation of five recognized *PHOX2A* mutations leading to CFEOM.

**Figure 6 children-09-01605-f006:**
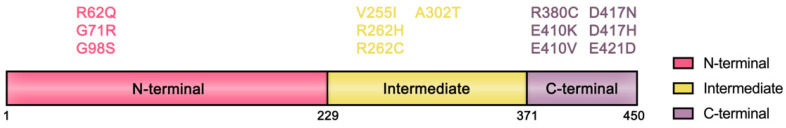
Schematic representation of 13 previously recognized TUBB3 amino acid substitutions associated with CFEOM.

**Table 1 children-09-01605-t001:** Clinical findings of CFEOM classified by genotype.

Gene/Locus	Inheritance	Penetrance	Phenotype	Featured Genotype–Phenotype Correlations
*KIF21A*	Autosomal dominant	Complete	CFEOM-1 (common) or CFEOM-3 (rare)	Large-angle exo-hypotropia Marcus Gunn jaw winking
*PHOX2A*	Autosomal recessive	Complete	CFEOM-2	Large-angle exotropia Pupil anomalies
*TUBB3*	Autosomal dominant	Incomplete	CFEOM-1 (rare) or CFEOM-3 (common)	Malformations of cortical development Cognitive impairment Significant phenotypic variability E410K syndrome R262H syndrome
*TUB* *B2B*	Autosomal dominant	Complete	CFEOM-3	Polymicrogyria Cognitive impairment
*TUBA1A*	Autosomal dominant	Unknown	CFEOM-1 or CFEOM-3	Polymicrogyria Cognitive impairment
*ECEL1*	Autosomal recessive	Incomplete	Undefinable	Arthrogryposis multiplex congenita
*COL25A1*	Autosomal recessive	Unknown	CFEOM-5	Exotropia Globe retraction with synergistic divergence Arthrogryposis multiplex congenita
Locus on chromosome 21qter	Autosomal recessive	Unknown	CFEOM-4 (Tukel syndrome)	Hand oligodactyly

## Data Availability

Not applicable.

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
