# Peer review of "Congenital Fibrosis of the Extraocular Muscles: An Overview from Genetics to Management"

_children, 2022, doi:10.3390/children9111605_

Round 1
Reviewer 1 Report
CFEOM is a tricky subject for a review article, because there is now so much literature to consider. The authors are to be commended for integrating fundamental science insights on genetics and pathophysiology with clinical approaches to vision, eye movement and eyelid management. The permitted word count would have been a limiting factor for such a broad review, and the authors manage to provide a pragmatic guide to the topic for clinicians, as well as a recommendation for further resarch at the end.
My only recommendation is that a review of the English language might be helpful, as the tenses sometimes change, and as it sometimes appears as if the authors had copied the original words from a paper to summarise findings, rather than paraphrasing the reports from other authors. This hinders the flow of reading. An example is "Recently, Dr. Khan does have raised an interesting point [67]". Throughout most of the manuscript, the references to not use the first author's name, and in addition, this sentence is grammatically wrong. In the same paragraph, most clinicians would not use the adjective "treacherous" in a scientific paper when mentioning malignant hyperthermia. In the following paragraph, the content of the sentence about referral of children at risk of amblyopia to an ophthlamologist is very important, but they only need to be referred once, and they the relevant eye care professional will provide regular review throughout the critical period of vision development.
Author Response
Dear Editors and Reviewers:
Thank you for your precious comments concerning our manuscript entitled "Congenital fibrosis of the extraocular muscles: An overview from genetics to management" (children-1908540). Those comments are all valuable and helpful for revising and improving our paper. We have studied the comments carefully and have made relevant corrections. Revised portions are highlighted in the manuscript. The main corrections in the manuscript and the point-by-point responses to your comments are as follows.
Reviewer 1
Critique 1. CFEOM is a tricky subject for a review article, because there is now so much literature to consider. The authors are to be commended for integrating fundamental science insights on genetics and pathophysiology with clinical approaches to vision, eye movement and eyelid management. The permitted word count would have been a limiting factor for such a broad review, and the authors manage to provide a pragmatic guide to the topic for clinicians, as well as a recommendation for further research at the end.
Response: We are grateful for the reviewer’s general comments and recognition of our work.
Critique 2. My only recommendation is that a review of the English language might be helpful, as the tenses sometimes change, and as it sometimes appears as if the authors had copied the original words from a paper to summarize findings, rather than paraphrasing the reports from other authors. This hinders the flow of reading. An example is "Recently, Dr. Khan does have raised an interesting point [67]". Throughout most of the manuscript, the references to not use the first author's name, and in addition, this sentence is grammatically wrong. In the same paragraph, most clinicians would not use the adjective "treacherous" in a scientific paper when mentioning malignant hyperthermia. In the following paragraph, the content of the sentence about referral of children at risk of amblyopia to an ophthlamologist is very important, but they only need to be referred once, and they the relevant eye care professional will provide regular review throughout the critical period of vision development.
Response: As requested, our manuscript has undergone English language editing by MDPI. We appreciate the reviewer for pointing out the issue on amblyopia surveillance, and the corresponding content was changed in the following:
Line 284: Younger patients at risk of amblyopia are extraly referred to eye care professionals throughout the critical period of vision development three or four times annually.

Reviewer 2 Report
The authors present a review of CFEOM and propose changes in how it is classified. It is not really clear who the target audience of this review is or what it adds to the literature, given that there have been several recent reviews on this topic. The discussion is superficial and the references cited, particularly in the introduction, are not the seminal papers on these topics.
The authors state that they are summarizing genotype-phenotype correlations, but do not address at all the exquisite genotype-phenotype correlates seen with TUBB3 variants. They do not address the 2 specific syndromes seen with the E410K and R262H TUBB3 amino acid substitutions. They repeatedly state that the TUBB3 phenotype is variable, which it is if you group all TUBB3 variants together, but if you look at each variant, the phenotype is quite standard.
The phenotypes for ECEL1 and COL25A1 need to be discussed in much more detail, as several of the reported patients would not be classified as CFEOM.
There are no clinical figures to display the phenotypes. The figures provided do not add much, as they are all only box plots of where reported variants are, they don’t even map variants to the protein structure. If the target audience of this review is clinicians, clinical photographs must be included.
It is not clear what new classification scheme for CFEOM the authors are proposing, or why that would be preferable to genetic classification. The field is moving towards purely genetic classification: CFEOM1 in patients with KIF21A variants, CFEOM2 with PHOX2A variants, and CFEOM3 with tubulin variants.
Specific issues:
Line 28:the chin-up posture is not just to “see-under” the ptotic lids, often the eyes cannot get to vertical midline and are stuck in infraduction
Line 63-64: CFEOM2 is higher prevalence in middle east due to consanguineous marriage and higher rates of autosomal recessive disease
Line 73-74: this sentence does not make grammatical sense
Table 1: Genotype column should be labeled phenotype; all have possibility of parental germline mosaicism (which has been shown with TUBB3); TUBB3 penetrance is incomplete for certain variants, complete for other variants; for both TUBB2B and TUBA1A, only certain variants give the CFEOM phenotype, other variants cause brain malformations without eye movement disorders; oculomotor synkinesis is possible with all forms of CFEOM1, it is not specific to CFEOM1
Line 98: “ocular motor neuron” rather than lower motor neuron
Line 135: TUBB3 is not indispensable. See Latremoliere et al Cell Reports 2018
Need to make clear that the variability depends on the exact amino acid substitution.
Line 152: G98S is one of the substitutions that causes both
Line 155: this is incorrect. Monocular elevation deficiency IS CFEOM, Moebius syndrome is sometimes misdiagnosed in patients with TUBB3 (if they have a TUBB3 variant, they do not have Moebius syndrome)
Line 234: many TUBB3 are de novo
Lines 236-240: English needs revision
Line 278: Heidary et al are not aggressive towards eyelid surgery. The issue with eyelid surgery is not the timing, but rather the final position of the lid.
Line 320: where is the reference for the reference table
In several cases, the wrong reference is cited.
Author Response
Dear Editors and Reviewers:
Thank you for your precious comments concerning our manuscript entitled "Congenital fibrosis of the extraocular muscles: An overview from genetics to management" (children-1908540). Those comments are all valuable and helpful for revising and improving our paper. We have studied the comments carefully and have made relevant corrections. Revised portions are highlighted in the manuscript. The main corrections in the manuscript and the point-by-point responses to your comments are as follows.
Reviewer 2
Critique 1. The authors present a review of CFEOM and propose changes in how it is classified. It is not really clear who the target audience of this review is or what it adds to the literature, given that there have been several recent reviews on this topic. The discussion is superficial and the references cited, particularly in the introduction, are not the seminal papers on these topics.
Response: We are grateful for the reviewer’s general comments and criticism. Actually, all of our authors are pediatric ophthalmologists. When a CFEOM patient is referred to us, we are consulted by parents about the diagnosis and prognosis (e.g. “Is there any other systemic abnormalities?”), etiology and inheritance (e.g. “What is the incidence if we had another baby?”), and management (e.g. strabismus, ptosis and amblyopia). Therefore, from our point of view, pediatric ophthalmologists may be the major target audience of this article. A summary on molecular genetic findings and genotype–phenotype correlations can help pediatric ophthalmologists better explain to parent. Meanwhile, a summary on management of can help them optimize the treatment plan.
We have adjusted the some references in the Introduction section. Still, the number of articles of CFEOM is relatively small by now (e.g. 164 results if you search “congenital fibrosis of the extraocular muscles” in Pubmed). We try to include all the up-to-date references as well as seminal papers. (e.g. retinal dysfunction were used to considered a feature of CFEOM-2, but it’s also recently demonstrated in CFEOM-1&3 in 2021, reference 4). Although there have been several recent reviews on this topic, CFEOM was still grouped into three subtypes in those classic seminal papers. To the best of our knowledge, this article is the first to summarize all the five proposed CFEOM phenotypes and the disease-causing genes, which we believe is systematic and meaningful. Hopefully this response can satisfy the reviewer.
Critique 2. The authors state that they are summarizing genotype-phenotype correlations, but do not address at all the exquisite genotype-phenotype correlates seen with TUBB3 variants. They do not address the 2 specific syndromes seen with the E410K and R262H TUBB3 amino acid substitutions. They repeatedly state that the TUBB3 phenotype is variable, which it is if you group all TUBB3 variants together, but if you look at each variant, the phenotype is quite standard.
Response: We thank the reviewer’s meaningful suggestion. TUBB3 E410K and R262H syndrome are two clinically significant disorders which demonstrating clinical genotype-phenotype correlation. We have added discussion related to these two specific mutations at Line 176: In particular, there are two distinct clinical disorders caused by two specific mutations, E410K and R262H, which can be recognized by their respective hallmarks…….
Critique 3. The phenotypes for ECEL1 and COL25A1 need to be discussed in much more detail, as several of the reported patients would not be classified as CFEOM.
Response: Thank you for the thoughtful comments. As suggested, we have added some details of CFEOM patients harboring ECEL1 or COL25A1 mutations. Meanwhile, some should not be classified as CFEOM were excluded. (Line 223 and Line 239)
Critique 4. There are no clinical figures to display the phenotypes. The figures provided do not add much, as they are all only box plots of where reported variants are, they don’t even map variants to the protein structure. If the target audience of this review is clinicians, clinical photographs must be included.
Response: We thank the review’s kind suggestion. We have added clinical photographs of patients with CFEOM-1, 2 and 3, which increases the readability of the article.
Critique 5. It is not clear what new classification scheme for CFEOM the authors are proposing, or why that would be preferable to genetic classification. The field is moving towards purely genetic classification: CFEOM1 in patients with KIF21A variants, CFEOM2 with PHOX2A variants, and CFEOM3 with tubulin variants.
Response: We thank the reviewer’s constructive suggestion, which does enhance the depth of the review. As suggest, we have added the classification method based on the genotype.
Line 411: One of the possible solutions is to classify CFEOM (defined as “congenital nonprogressive restrictive ophthalmoplegia with or without ptosis) into six types purely according to genotype, regardless of the clinical manifestations. To be specific ……
Specific issues:
Critique 6. Line 28:the chin-up posture is not just to “see-under” the ptotic lids, often the eyes cannot get to vertical midline and are stuck in infraduction
Response: We appreciate the reviewer for pointing out this issue. Besides a chin-up posture, head tilt is also not uncommon. The corresponding text was modified to make the context more scientific.
Critique 7. Line 63-64: CFEOM2 is higher prevalence in middle east due to consanguineous marriage and higher rates of autosomal recessive disease.
Response: Thanks for your helpful comment. The corresponding text was changed accordingly.
Line 26: The most distinctive feature of ophthalmoplegia is that vertical eye movements are severely restricted, commonly resulting in a compensatory chin-up head position or head tilt.
Critique 8. Line 73-74: this sentence does not make grammatical sense.
Response: We thank the reviewer for pointing out our grammatical errors. We have checked all the potential writing errors and polished our manuscript with the help of MDPI English language editing service.
Critique 9. Table 1: Genotype column should be labeled phenotype; all have possibility of parental germline mosaicism (which has been shown with TUBB3); TUBB3 penetrance is incomplete for certain variants, complete for other variants; for both TUBB2B and TUBA1A, only certain variants give the CFEOM phenotype, other variants cause brain malformations without eye movement disorders; oculomotor synkinesis is possible with all forms of CFEOM1, it is not specific to CFEOM1
Response: We appreciate a lot for the review’s kind and careful comments, which are all valuable and helpful for revising and improving our paper. We have made relevant corrections as suggested:
Table 1: “Genotype” corrected to “Phenotype”;
Table 1: deletion of “Possibility of parental germline mosaicism”;
Line 163: TUBB3 penetrance: adding “with some displaying incomplete penetrance (e.g. R262H and R262C)”;
Table 1: deletion of “Malformations of cortical development” in both the TUBB2B and TUBA1A Row;
Line 46: Deletion of “Aberrant convergent or divergent eye movements and”
Critique 10. Line 98: “ocular motor neuron” rather than lower motor neuron.
Response: Thanks for your helpful comment. The corresponding text was changed accordingly.
Critique 11. Line 135: TUBB3 is not indispensable. See Latremoliere et al Cell Reports 2018
Need to make clear that the variability depends on the exact amino acid substitution.
Response: Thank you for letting us know about this significant reference. We have added this reference and modified the relevant content.
Critique 12. Line 152: G98S is one of the substitutions that causes both.
Response: Thank you for the thoughtful comments. The point we are making is that G98S some causes both of CFEOM and MCD, but sometimes cause MCD only. We have modified the sentence to make it more clear for readers.
Line 170: Another substitution (G98S), previously known to cause CFEOM-3, was identified in an MCD pedigree, who, however, did not meet the diagnostic criteria for COFEM.
Critique 13. Line 155: this is incorrect. Monocular elevation deficiency IS CFEOM, Moebius syndrome is sometimes misdiagnosed in patients with TUBB3 (if they have a TUBB3 variant, they do not have Moebius syndrome).
Response: Thank you for pointing our mistakes. The relevant content has been revised.
Line 172: Interestingly, recent studies demonstrated that infantile nystagmus could arise without CFEOM owing to TUBB3 variant.
Critique 14. Line 234: many TUBB3 are de novo.
Response: We are grateful for this important advice, which makes our article more rigorous.
Line 273: For example, a patient harboring TUBB3 mutation may have unremakable family history because of incomplete penetrance or de novo mutation
Critique 15. Lines 236-240: English needs revision
Response: We thank the reviewer for pointing out our grammatical errors. We have checked all the potential errors.
Critique 16. Line 278: Heidary et al are not aggressive towards eyelid surgery. The issue with eyelid surgery is not the timing, but rather the final position of the lid.
Response: We appreciate the reviewer’s comment. We have deleted the relevant description of “aggressive attitude of Heidary” at Line 319.
Critique 17. Line 320: where is the reference for the reference table
Response: We are grateful for this important advice. Since this is a general surgical table for congenital ptosis, we described it as the “recommended reference table”. Certainly, it seems not rigorous enough in the setting of CFEOM. Therefore, we have deleted the relevant content. Thank you again for improving the scientificity of our article.
Critique 18. In several cases, the wrong reference is cited.
Response: We appreciate the reviewer for pointing out this issue. We have checked all the potential errors.
